# Examining the Impact of COVID-19 on Entrepreneurial Intention through a Stimulus–Organism–Response Perspective

**Gentjan Çera** [1,*], **Margarita Ndoka** [2], **Ines Dika** [3] **and Edmond Çera** [4]

1    Faculty of Economics and Agribusiness, Agricultural University of Tirana, 1001 Tirana, Albania
2    Faculty of Economy, European University of Tirana, 1001 Tirana, Albania
3    Faculty of Economics, University of Tirana, 1001 Tirana, Albania
4    Faculty of Management and Economics, Tomas Bata University in Zlin, 760 01 Zlín, Czech Republic
*    Correspondence: gcera@ubt.edu.al

**Abstract:** Among scholars, there is an interest in understanding how entrepreneurial behavior is influenced by the consequences of crises. The COVID-19 pandemic may negatively or positively affect individuals' behavior, including entrepreneurial intention. Thus, this paper seeks to study whether or not the economic shock caused by the pandemic reinforces the intention to start a business. The research was administered at the individual level by distributing a structured survey. The hypotheses were developed based on a unique conceptual framework integrating the planned behavior theory and a stimulus–organism–response perspective. The relationships were tested using the structural equation modeling method with an original dataset of more than 800 respondents from three post-communist transition countries. The results indicate that the COVID-19 pandemic, seen as an opportunity, positively influences both the antecedents of entrepreneurial intention and individuals' intention to start a business. The message that these findings convey is that, even in crises, there are opportunities from which one can benefit, including the individual's propensity to engage in startup activities. By examining the impact of the COVID-19 crisis on entrepreneurial behavior, educational institutions and policymakers can design effective policies to foster entrepreneurship and reduce unemployment, particularly among the youth.

**Keywords:** COVID-19; entrepreneurial intention; PLS-SEM; theory of planned behavior; Albania; Kosovo; North Macedonia

## 1. Introduction

It is generally accepted among scholars that disasters and crises lead to economic and societal changes in people's behavior and lifestyles (Menter 2022; Rayburn et al. 2022). Such changes can manifest as negative and positive influences on entrepreneurial activity (Krichen and Chaabouni 2021; Meahjohn and Persad 2020). Therefore, an exogenous shock not only poses additional challenges to individuals, organizations, and economies, but can also offer them new opportunities for business innovation (Brown and Rocha 2020). According to Aly (2022), entrepreneurship is seen as a vital factor in achieving a resilient economy in times of crisis. Entrepreneurial activity can be fed by encouraging and motivating individuals to create new businesses. Prior research has shown that in order to avoid failure and to ensure sustainability, individuals and organizations must be provided with support during crises (Noelia and Rosalia 2020; Ratinho et al. 2020; Çera et al. 2019; Dvorský et al. 2019; Alshebami and Seraj 2022b).

The COVID-19 pandemic is an unprecedented event that spread quickly worldwide. Being a highly infectious illness, it has impacted global public health because of its high level of transmission and increased death rate—mostly among the elderly, people with impaired immune systems, and those with underlying medical conditions (Mueller et al. 2020). Today, even though most of the governmental measures have been removed globally,

the infection is still present (Our World in Data n.d.). This crisis has definitely changed the behavior in terms of how individuals work and live (Hale et al. 2021; Ratten 2021).

Generally, practitioners and academics believe that fostering entrepreneurship in times of crisis and economic recession is an adequate response (Capella-Peris et al. 2020; Meahjohn and Persad 2020). The COVID-19 pandemic has threatened public health by putting it under pressure and forcing governments to implement measures such as lockdowns. Nevertheless, this pandemic has created new opportunities for entrepreneurs (Ketchen and Craighead 2020; McGee and Terry 2022; Usman and Sun 2022), and this may represent the right moment for individuals who want to carry on their career in entrepreneurship (Godswill et al. 2021; Krichen and Chaabouni 2021; Ruiz-Rosa et al. 2020).

Considering the benefits provided by entrepreneurial activity—including social and economic aspects (decreasing the unemployment rate), especially for young adults—researchers, educational institutions, and public officials (i.e., governments) are particularly interested in having a better view of the impact of various factors on individuals' entrepreneurial behavior, including the intention to start a business. Such interest is more present in times of crisis, including the COVID-19 pandemic. A better understanding of these determinants (particularly during a crisis) would make it possible to design new policies or adjust existing ones to boost entrepreneurial activity.

According to Ratten (2021), the pandemic should be seen not only as a cause of considerable havoc, but also as a crisis that created an environment suitable for new entrepreneurial opportunities to flourish. Hence, the adversity of COVID-19 may lead to a new way of doing business (Usman and Sun 2022). Therefore, it would be interesting to see the actual effect of the COVID-19 crisis on individuals' intention to start a business.

Even though there are a considerable number of papers covering entrepreneurial intention (Abebe and Alvarado 2018; Barba-Sánchez and Atienza-Sahuquillo 2018; Belas et al. 2017; Neneh 2019; Palalić et al. 2017; Perez-Quintana et al. 2017; Zarnadze et al. 2022; Çera et al. 2021), minimal research has focused on the role of the COVID-19 pandemic on increasing individuals' intention to start up a business (Godswill et al. 2021; Hernández-Sánchez et al. 2020; Li et al. 2022; Ratten 2021; Trif et al. 2022). Therefore, this paper seeks to shed light on the relationship mentioned above by introducing an integration of two theories: the theory of planned behavior (Ajzen 1991), and the stimulus–organism–response perspective (Mehrabian and Russell 1974). Such research will provide useful insights for the entrepreneurship literature and policymakers.

The rest of this paper is organized as follows: The article's next section is dedicated to theoretical lenses and the development of hypotheses. Then, the results are interpreted after the description of the methodological procedures. The fifth section of the article consists of a discussion of the findings, followed by the section dedicated to the conclusion.

## 2. Literature Review

### 2.1. Theoretical Lenses

The present study uses two theoretical lenses: the theory of planned behavior (Ajzen 1991), and a stimulus–organism–response framework (Mehrabian and Russell 1974). The literature on these theoretical views in the context of entrepreneurial intention is discussed below.

Scholars consider individuals' intentions towards startups to be a difficult topic to study (Liñán and Fayolle 2015; Maheshwari et al. 2022). The complexity of this topic lies in the fact that individuals' intention is affected by several factors (Shane et al. 2003; Murnieks et al. 2020; Lüthje and Franke 2003), including the mental process that underlies the intentional actions (Entrialgo and Iglesias 2020) and the sophisticated process based on perception (Krueger and Carsrud 1993; Krueger et al. 2000). One of the predominant models used to study this topic is the theory of planned behavior (Maheshwari et al. 2022), introduced by Ajzen (1991), which proposes that attitudes, subjective norms, and perceived behavioral control are three key determining factors of one's intention towards a particular action and, in turn, leading to that person's actual action or behavior. The efficacy of this theory has been tested, showing that the model works (Krueger and Carsrud 1993; Kautonen et al. 2015; Munir et al. 2019;

van Gelderen et al. 2008; Zampetakis et al. 2017). The majority of the papers that used this theory applied the model without the relationship between intention and action/behavior. However, there is evidence of a strong correlation between an individual's intention and their actual behavior toward starting a business (Neneh 2019). In a meta-analysis, Armitage and Conner (2001) found that the intention–behavior correlation was statistically significant, reflecting a medium-sized effect ($r = 0.47$). Therefore, studying entrepreneurial intention may provide insights into the actual behavior towards starting a business. Moreover, this model has been used in the context of the COVID-19 pandemic (Ruiz-Rosa et al. 2020; Godswill et al. 2021; Krichen and Chaabouni 2021).

As mentioned earlier, in this paper, a different theory is applied that complies with the theory of planned behavior: the stimulus–organism–response perspective. This theory was introduced by Mehrabian and Russell (1974), consisting of three elements: stimulus, organism, and response. In this framework, stimuli refer to a set of factors, including the environment and information load. The organism is the second element of this framework, and it refers to the organism's conditions, which consist of emotional reactions to environmental stimuli. The third and final element of this framework is labeled as "response", which represents an approach or avoidance action or behavior.

These two theoretical perspectives can be merged to provide a better view of the context of the present study. Hence, the COVID-19 pandemic is seen as a stimulus coming from the external environment, affecting an individual's organism conditions. In this study, the organism is represented by determinants of entrepreneurial intention (i.e., attitude, subjective norms, and perceived behavioral control). Lastly, entrepreneurial intention covers the response component of the stimulus–organism–response perspective.

*2.2. Development of Hypotheses*

2.2.1. Attitudes towards Behavior and Entrepreneurial Intention

Once the theoretical lenses used in this study were set, the development of the hypothesis could proceed. The following paragraphs discuss the relationships based on the two mentioned theories. The first four hypotheses deal with the theory of planned behavior, while the last set represents the relationships between COVID-19 and other factors.

An individual's attitude towards entrepreneurship is defined as the extent to which a person holds a negative or positive attitude towards becoming an entrepreneur (Liñán and Chen 2009). From this definition, one can say that people with a positive perception of being an entrepreneur are more likely to have a firm interest in engaging in startup activity, whereas people with a negative perception are more likely to have no interest in such activity. Prior research demonstrates that there is a positive association between attitude and entrepreneurial intention (Joensuu-Salo et al. 2015; Feola et al. 2019; Maes et al. 2014; Haus et al. 2013; Liñán and Chen 2009), including limited research covering the time of the COVID-19 pandemic (Ruiz-Rosa et al. 2020). Nevertheless, some studies do not report a significant influence of attitudes on entrepreneurial intention, even during COVID-19 (Godswill et al. 2021; Nguyen et al. 2020). Thus, it is not clear whether attitude's effect on entrepreneurial intention is positive. Therefore, there is a need to study this relationship. Thus, our first hypothesis is as follows:

**Hypothesis 1 (H1).** *Personal attitude towards entrepreneurship positively influences entrepreneurial intention.*

2.2.2. Subjective Norms and Entrepreneurial Intention

According to the theory of planned behavior, the second determinant of a person's intention is the subjective norm, which is known as the social influence on an individual to perform (or not) a particular behavior (Ajzen 1991). This is related to the belief that an important person, relatives, friends, or others will endorse (or not) a specific behavior, e.g., a decision to start up a business. Prior studies show a positive effect of subjective norms on entrepreneurial intention (Moriano et al. 2012; Rantanen and Toikko 2017; Mirjana et al.



2018; Maresch et al. 2016; Misoska et al. 2016). Moreover, it is difficult to find a paper reporting an insignificant relationship—for example, the study of Godswill et al. (2021), which was conducted in the context of the COVID-19 pandemic. The present study may offer additional evidence about this relationship in the context of the pandemic. Thus, subjective norms (i.e., social influence) are expected to positively predict one's intention to start a business. Therefore, our second hypothesis is as follows:

**Hypothesis 2 (H2).** *An individual's entrepreneurial intention is positively influenced by subjective norms.*

### 2.2.3. Perceived Behavioral Control and Entrepreneurial Intention

Based on the theory of planned behavior, perceived behavioral control is the third main determinant of an individual's intention (Ajzen 1991). In the context of entrepreneurship, this is seen as the belief and confidence that a person has in carrying out business activities as an entrepreneur. Based on this logic, the more opportunities and resources a person believes they have and the fewer constraints they foresee, the greater their perceived control over a particular action is expected to be, including startup activity. Previous studies confirm the positive effect of perceived behavioral control on entrepreneurial intention (Al-Jubari 2019; Joensuu-Salo et al. 2015; Kautonen et al. 2015; Liñán and Chen 2009; Nguyen et al. 2020), including those conducted during the COVID-19 pandemic (Ruiz-Rosa et al. 2020; Godswill et al. 2021). Although there is such evidence, there is a need to study this relationship in the context of COVID-19 in post-communist countries. Thus, our third hypothesis is as follows:

**Hypothesis 3 (H3).** *Perceived behavioral control positively influences entrepreneurial intention.*

### 2.2.4. The Role of COVID-19

Previous studies have tried to shed light on the impact of COVID-19 on different aspects of entrepreneurship, including the intention to start a business (Lopes et al. 2021; Botezat et al. 2022). Arve et al. (2022) conducted an experiment and found that the majority of prospective entrepreneurs either canceled or postponed their projects during the first months of the pandemic. Nevertheless, some studies see this crisis as a chance to implement a business idea by establishing a firm. Research found that most of the students from Erasmus University Rotterdam did not change their entrepreneurial intention due to COVID-19 (Wismans et al. 2022). In addition, the latter study demonstrated that the share of students who increased their entrepreneurial intention (19%) was higher than those who decreased such intention (16%). Hence, evidence supports the claim that COVID-19 offers new chances for entrepreneurship. Moreover, seeing COVID-19 as an opportunity to engage in entrepreneurial activity is more common than perceiving it as a threat (Lungu et al. 2021). This finding is supported by a prior study conducted in a war setting, which suggests that even under conditions of war, people develop entrepreneurial intentions in case they can grow from adversity and believe in their abilities (Bullough et al. 2014). Thus, one can say that crisis may create a suitable environment for individuals to see entrepreneurial opportunities. According to Krichen and Chaabouni's (2021) research, there is a positive and statistically significant impact of COVID-19 seen as an opportunity on students' likelihood to start a business. This finding is consistent with other research that highlights the pandemic's potential beneficial effects on entrepreneurship (Botezat et al. 2022; Lungu et al. 2021). Consequently, a positive effect of COVID-19 on entrepreneurial intention was also expected to be present in this study.

Recently published papers have utilized the theory of planned behavior to explore the impact of COVID-19 on behavioral changes, including the effects of COVID-19 on the determinants of behavioral intention (i.e., attitude, subjective norms, and perceived behavioral control) (Srisathan and Naruetharadhol 2022; Prasetyo et al. 2020; Han et al. 2020; Lucarelli et al. 2020). It is generally known that external factors influence individuals' attitudes towards particular actions. In this context, according to Rayburn et al. (2022), in response to the COVID-19 pandemic, individuals moved from fear to frugality, either by following new behaviors forced by the crisis, or by going back to their behavior prior to the crisis. Hence, attitudes towards different aspects change in a crisis setting, such as attitudes towards entrepreneurship in general and starting up a business. In the context of the COVID-19 pandemic, Gomes et al. (2021) demonstrated that the positive and significant influence of attitudes toward behavior and entrepreneurial intention was present in both situations: before and during the pandemic. Moreover, the latter study shows a slightly more significant effect during the COVID-19 pandemic than before it.

Similar to attitudes, evidence shows that subjective norms and perceived behavioral control increased due to COVID-19 (Botezat et al. 2022). According to prior research, people's lifestyles have changed due to COVID-19 (Rayburn et al. 2022; Ratten 2021). At the community level, to avoid the transmission of illness, individuals were recommended to take additional hygienic measures. Individuals are pursuing digitization more aggressively than ever before in order to respect social distancing norms, embracing new activities and interactions—including teleworking—and adjusting everyday habits to fit a new reality (Srisathan and Naruetharadhol 2022). Therefore, a person's friends and relatives may push them to take action to start a business, meaning that subjective norms are influenced by COVID-19. Indeed, previous research supports such an association (Prasetyo et al. 2020; Srisathan and Naruetharadhol 2022; Han et al. 2020).

Very few papers have discussed the impact of COVID-19 on perceived behavioral control. By definition, perceived behavioral control is the comfort level of a person in performing any particular behavior (Ajzen 1991). Its determinants are assumed to be the set of accessible control beliefs, such as beliefs about the presence of factors that can enable or constrain a certain behavior. This reasoning leads to the concept of resilience, which refers to the ability that a person has to recover from or adjust easily to change or misfortune (Sinclair and Wallston 2004; Alshebami and Seraj 2022a). Studies have shown that resilience is an important factor in crisis settings, including in entrepreneurship (Arve et al. 2022; Bullough et al. 2014; Sharma and Rautela 2021; Schepers et al. 2021; Alshebami 2022). Prior research has found that perceived behavioral control is affected by crises, including COVID-19, supporting the existence of this association (Prasetyo et al. 2020; Srisathan and Naruetharadhol 2022).

Based on the above discussion, one can conclude that COVID-19 influences attitudes toward entrepreneurship, subjective norms, and perceived behavioral control. Thus, our fourth hypothesis is as follows:

**Hypothesis 4a–c (H4a–c).** *The COVID-19 pandemic has a positive effect on attitudes to start a business (H4a), subjective norms (H4b), and perceived behavioral control (H4c).*

**Hypothesis 4d (H4d).** *Entrepreneurial intention is positively affected by the COVID-19 pandemic.*

The integration of the theory of planned behavior and the stimulus–organism–response perspective is illustrated in Figure 1. Additionally, the figure also shows the proposed linkages (i.e., hypotheses).

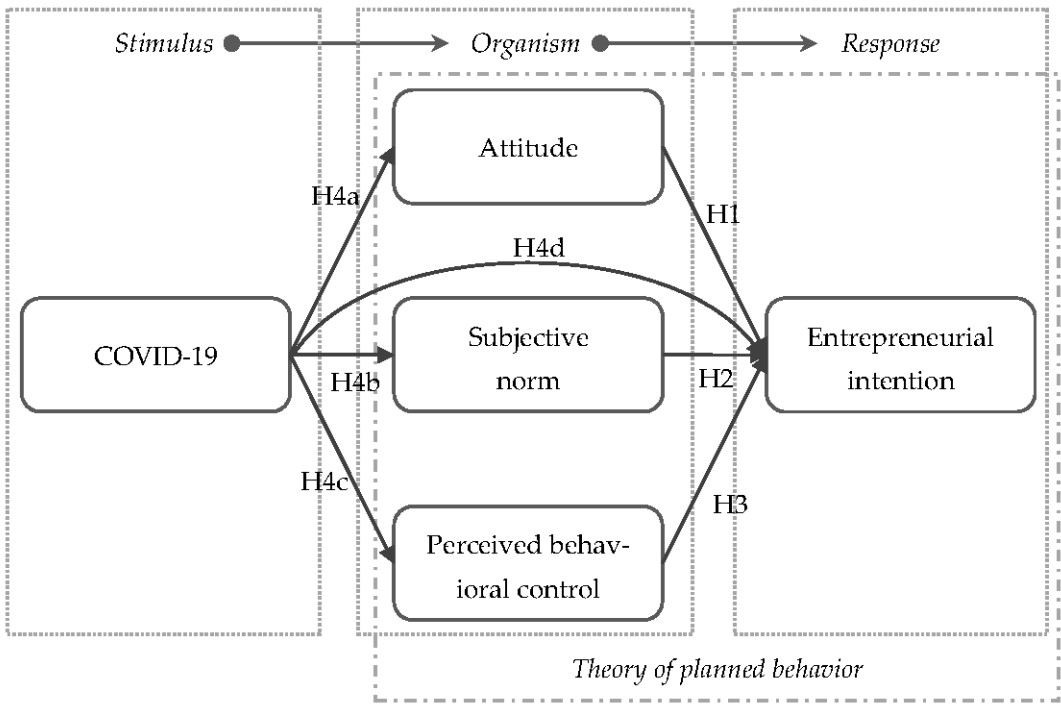

**Figure 1.** Conceptual framework and hypotheses.

### 3. Method and Procedures

*3.1. Research Instrument and Sample*

In order to meet the goals of this research, a survey was conducted to test the research model and indicate the significance of the relationships. The use of surveys is a quantitative method that can infer the population by studying a sample (Creswell and Creswell 2017). This type of method implies the need for primary data collection. Hence, a questionnaire was developed based on the literature review.

The research covered three countries: Albania, Kosovo, and North Macedonia. After the validation of the questionnaire, it was translated into the Albanian and Macedonian languages. The data were collected during the COVID-19 pandemic at the end of 2021.

The respondents were selected by following a two-stage sampling procedure: (i) selection of primary sampling unit, and (ii) selection of the respondents. The first stage was fulfilled by randomly selecting participants from among the voting centers. The second stage consisted of selecting the respondents following a methodology of starting from the voting center and then moving clockwise, always getting further from the starting point. More than 800 valid responses were collected, with more than 200 respondents from each country. Such a sample size is well above the recommendation of Hair et al. (2010).

Table 1 shows the sample profile (overall and per country). For the most part, the pattern of the subsample profiles reflects one of the overall samples. Three out of five respondents were 24 years old or less. The majority of the respondents were female. Almost 70% of the respondents were settled in urban areas (i.e., cities).

*3.2. Measurement of Variables*

The variables of this research were measured as proposed in the literature, with minor changes, including wording or adaptation to the context. The dependent variable in this paper is entrepreneurial intention. There are different ways in which this variable has been measured in the literature (Armitage and Conner 2001; Çera and Çera 2020; Franke and Lüthje 2004; Krueger and Carsrud 1993; Lim et al. 2016; Çera et al. 2020). However, as claimed by Thompson (2009), an individual's intention cannot be captured by considering only one item/statement; therefore, entrepreneurial intention in this work is measured

by four items/statements, which can be found in the Appendix A. The source for this measurement was the work published by Liñán and Chen (2006).

**Table 1.** Sample profile.

| Variable | Category | Country | | | |
|---|---|---|---|---|---|
| | | **Albania**<br>**n = 412** | **Kosovo**<br>**n = 207** | **North Macedonia**<br>**n = 203** | **Total**<br>**N = 822** |
| Settlement | City | 87.9% | 48.8% | 49.3% | 68.5% |
| | Village | 12.1% | 51.2% | 50.7% | 31.5% |
| | Total | 100% | 100% | 100% | 100% |
| Gender | Male | 26.7% | 29.0% | 25.1% | 26.9% |
| | Female | 73.3% | 71.0% | 74.9% | 73.1% |
| | Total | 100% | 100% | 100% | 100% |
| Age | 18–24 years old | 66.5% | 46.4% | 58.1% | 59.4% |
| | 25–35 years old | 33.5% | 53.6% | 41.9% | 40.6% |
| | Total | 100% | 100% | 100% | 100% |

Regarding the independent variables, excluding the COVID-19 variable, all of the others were measured similarly to the approach of García-Rodríguez et al. (2017). A single-item variable was used to measure the impact of COVID-19 on the antecedents of the individuals' intent to act and their intentions themselves. The statement reads "the COVID-19 pandemic situation has made me optimistic about starting a business". The respondents were asked to indicate their level of agreement with the statement (1 = strongly disagree, 5 = strongly agree). A similar type of measurement was used in a prior study (Krichen and Chaabouni 2021). Appendix A (Table A1) summarizes the list of items/indicators used to measure each variable included in this research.

### 3.3. Method

The partial least squares structural equation modelling (PLS-SEM) method was used to test the proposed conceptual framework. PLS-SEM was performed using SmartPLS 3.0 (Ringle et al. 2015) computer software. The PLS approach is a variance-based structural equation modeling (SEM) method (Hair et al. 2017). This approach enables assessment of the measurement model, including the reliability and validity of the constructs and the structural model. Therefore, it can test the formulated hypotheses by examining the standardized path coefficients. As recommended by the literature, the standardized coefficients were estimated using the bootstrap procedure, with 5000 iterations of resampling (Hair et al. 2019).

Since the three countries share similar cultures and levels of economic development, our analysis considered one dataset rather than three sub-datasets (one per country). According to Hofstede (2011), these countries share very similar cultural values (see Figure 2). Unfortunately, there are no reports for Kosovo. However, Kosovo is inhabited by Albanians and has many things in common not only with Albania, but also with North Macedonia. As the graph depicts, there are few differences between Albania and North Macedonia. Therefore, the three countries share similar cultural values. This leads to the suggestion of analyzing the data as a whole, rather than separately.

### 3.4. Checking Assumptions

A PLS-SEM method is an approach based on assumptions. Their violation (individually or collectively) leads to problems in the interpretation of the results that this method generates. Therefore, the violation of any of this approach's assumptions is an indication that its output is misleading. To avoid such issues there is a need to check some assumptions, which are mostly related to the measurement model, including the reliability and validity of the items and scales.

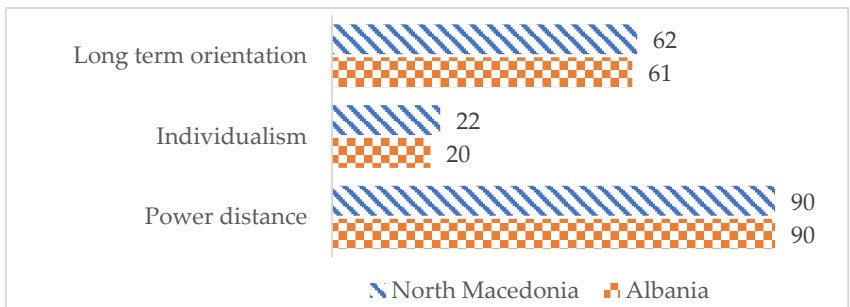

**Figure 2.** Hofstede's cultural dimensions for Albania and North Macedonia. *Source*: Hofstede Insights: https://www.hofstede-insights.com/ (accessed on 22 October 2022).

In order to assess the fitness of the model, a list of metrics can be examined. In this context, Cronbach's alpha, composite reliability (CR), and rho alpha provide information about scale reliability, while average variance extracted (AVE) reports the extent to which the scale reliability and convergent validity are satisfactory. These metrics are assessed and reported in Table 2. Since the values of Cronbach's alpha (above 0.70), composite reliability (above 0.60), and rho alpha are above the thresholds for all scales (Hair et al. 2019), it can be said that the data show satisfactory reliability and convergent validity of the constructs. In addition, item reliability can be assessed by examining the factor loadings, which should be above 0.708 (Hair et al. 2019). Indeed, as reported in Table 2, all loadings are above this threshold, leading to the conclusion that all constructs explain more than half of the indicator's variance, providing evidence to accept indicator reliability.

**Table 2.** Descriptive statistics and measurement model quality attributes.

| Variable | Mean | Standard Deviation | Loadings | VIF | CA | rho_A | CR | AVE |
|---|---|---|---|---|---|---|---|---|
| COVID-19 | 2.20 | 1.21 | 1 | 1 | 1 | 1 | 1 | 1 |
| EI | - | - | - | - | 0.9079 | 0.9104 | 0.9354 | 0.7837 |
| ei1 | 3.19 | 1.26 | 0.8639 | 2.4197 | | | | |
| ei2 | 3.25 | 1.22 | 0.8979 | 2.8824 | | | | |
| ei3 | 3.48 | 1.30 | 0.9018 | 3.1293 | | | | |
| ei4 | 3.50 | 1.28 | 0.8768 | 2.7662 | | | | |
| ATT | - | - | - | - | 0.9349 | 0.9357 | 0.9535 | 0.8367 |
| att1 | 3.29 | 1.27 | 0.8976 | 3.0892 | | | | |
| att2 | 3.40 | 1.32 | 0.9270 | 4.0227 | | | | |
| att3 | 3.56 | 1.35 | 0.9140 | 3.5009 | | | | |
| att5 | 3.28 | 1.30 | 0.9199 | 3.7092 | | | | |
| SN | - | - | - | - | 0.8739 | 0.8919 | 0.9215 | 0.7966 |
| sn1 | 3.74 | 1.22 | 0.8783 | 1.8999 | | | | |
| sn2 | 3.59 | 1.22 | 0.9223 | 3.4762 | | | | |
| sn3 | 3.33 | 1.21 | 0.8762 | 2.9185 | | | | |
| PBC | - | - | - | - | 0.9047 | 0.9067 | 0.9265 | 0.6777 |
| pbc1 | 3.62 | 1.18 | 0.8193 | 2.3296 | | | | |
| pbc2 | 3.53 | 1.11 | 0.8517 | 2.5354 | | | | |
| pbc3 | 3.68 | 1.14 | 0.8589 | 2.7262 | | | | |
| pbc5 | 3.33 | 1.17 | 0.7913 | 2.0059 | | | | |
| pbc6 | 3.37 | 1.12 | 0.7983 | 2.1293 | | | | |
| pbc7 | 3.28 | 1.12 | 0.8175 | 2.2221 | | | | |

*Note*: VIF, variance influence factor; CA, Cronbach's alpha; CR, composite reliability; AVE, average variance extracted; ATT, attitude; EI, entrepreneurial intention; PBC, perceived behavioral control; SN, subjective norms; COVID-19, the COVID-19 pandemic.

Moreover, Table 2 shows the variance influence factor (VIF) for each indicator. In general, VIF indicates the presence of multicollinearity in a relationship. However, since

the data show that the VIF values are below 5 (Hair et al. 2019), one can say that there is no multicollinearity issue within the measurement model.

Another crucial issue to consider in PLS-SEM deals with the discriminant validity, which indicates how distinct one construct is from others. Table 3 provides information on this issue, since it reports the correlations' heterotrait–monotrait ratio (HTMT). It is recommended to examine HTMT coefficients when using PLS-SEM as a measure of discriminant validity (Henseler et al. 2015). The rule of thumb is that the HTMT values should be below 0.85. In Table 3, all of the coefficients satisfy this rule. This test result indicates that the discriminant validity is set in this paper. Additionally, Table 3 reports the correlation coefficients among the measured constructs.

**Table 3.** Correlation matrix and discriminant validity—HTMT.

|  | ATT | COVID-19 | EI | PBC | SN |
|---|---|---|---|---|---|
| ATT | | 0.2598 | 0.5888 | 0.6382 | 0.4566 |
| COVID-19 | 0.2687 | | 0.2196 | 0.2225 | 0.1657 |
| EI | 0.6370 | 0.2297 | | 0.4711 | 0.3801 |
| PBC | 0.6918 | 0.2353 | 0.5186 | | 0.5778 |
| SN | 0.4936 | 0.1717 | 0.4172 | 0.6374 | |

*Note*: Correlation coefficients are above the diagonal, while HTMT coefficients are below it. ATT, attitude; EI, entrepreneurial intention; PBC, perceived behavioral control; SN, subjective norms; COVID-19, the COVID-19 pandemic.

Figure 3 graphically illustrates the main results of the measurement model, as generated by SmartPLS 3.0.

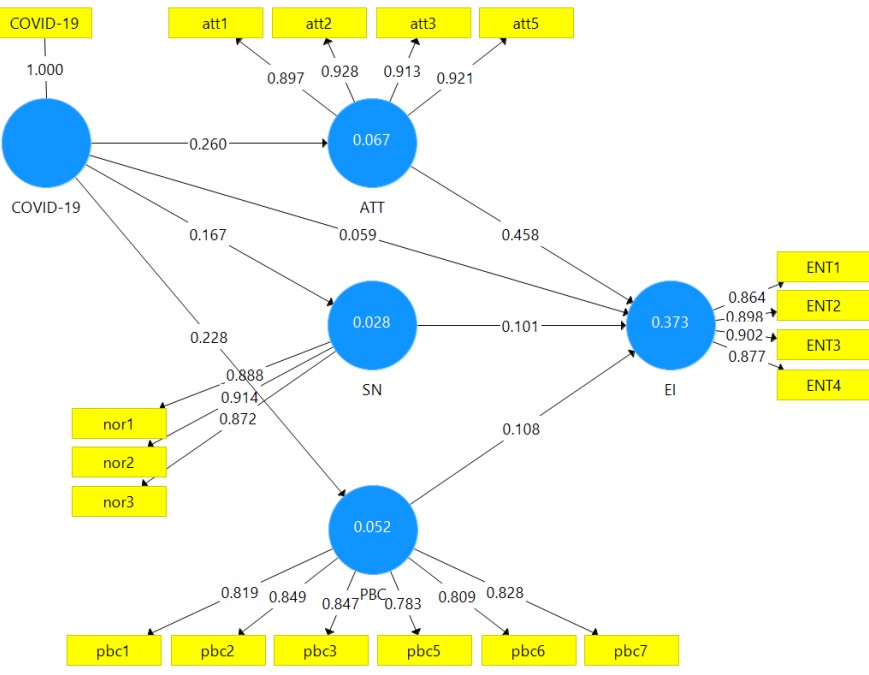

**Figure 3.** Measurement model. *Note*: ATT, attitude; EI, entrepreneurial intention; PBC, perceived behavioral control; SN, subjective norms; COVID-19, the COVID-19 pandemic.

## 4. Results

Upon checking the assumptions of the PLS-SEM method, the output of the analysis can be interpreted. This means that the satisfaction of the PLS-SEM's assumptions leads to the examination of the formulated hypotheses. The tested model explains 37.2% of the variation in entrepreneurship intention, 6.7% in attitude, 5.2% in perceived behavioral control, and almost 3% in subjective norms. These statistics are summarized in Table 4.

**Table 4.** R-squares.

| Construct | R Squared | Adjusted R Squared |
|---|---|---|
| Attitude | 0.067 | 0.066 |
| Entrepreneurial intention | 0.372 | 0.369 |
| Perceived behavioral control | 0.052 | 0.051 |
| Subjective norms | 0.028 | 0.027 |

According to the proposed conceptual framework, the entrepreneurial intention is determined by attitude, subjective norms, perceived behavioral control, and COVID-19. The results of the path analysis are summarized in Table 5. As indicated in the Method and Procedures section, the path coefficient's statistical significance was examined to conclude whether the hypotheses were supported or not.

**Table 5.** Results of hypotheses testing via bootstrapping (direct effect).

| Hypothesis | Path | Coefficient | *t*-Value | VIF |
|---|---|---|---|---|
| H1 | ATT → EI | 0.459 | 11.98 *** | 1.762 |
| H2 | SN → EI | 0.101 | 2.902 ** | 1.532 |
| H3 | PBC → EI | 0.108 | 2.672 ** | 2.052 |
| H4a | COVID-19 → ATT | 0.260 | 8.313 *** | 1.028 |
| H4b | COVID-19 → PBC | 0.228 | 6.952 *** | 1.028 |
| H4c | COVID-19 → SN | 0.165 | 5.108 *** | 1.000 |
| H4d | COVID-19 → EI | 0.220 [a] | 6.492 *** | 1.762 |

*Note*: VIF, variance influence factor; ATT, attitude; EI, entrepreneurial intention; PBC, perceived behavioral control; SN, subjective norms; COVID-19, the COVID-19 pandemic; a, total effect; ** and *** imply that the test result is significant at the 99% and 99.9% levels, respectively.

Subjective norms positively influenced attitudes ($\beta = 0.426$, $t = 13.68$, $p < 0.001$) and perceived behavioral control ($\beta = 0.557$, $t = 21.49$, $p < 0.001$). These findings support H1a and H1b, meaning that subjective norms are a significant determinant of both an individual's attitude and their perceived behavioral control. The data show that an individual's intention toward startups is statistically significantly and positively affected by attitude ($\beta = 0.459$, $t = 11.98$ $p < 0.001$), subjective norms ($\beta = 0.101$, $t = 2.902$, $p < 0.01$), and perceived behavioral control ($\beta = 0.108$, $t = 2.672$, $p < 0.01$). Thus, there is evidence in support of H1, H2, and H3. These hypotheses deal with the standard model of the theory of planned behavior (Ajzen 1991). The remaining hypotheses link the impact of the COVID-19 pandemic with the theory of planned behavior variables.

In this paper, the role of stimulus in the stimulus–organism–response paradigm is played by the COVID-19 pandemic, which influences all factors mentioned in the theory of planned behavior (see Figure 1). The data show that COVID-19 statistically and positively influences attitude ($\beta = 0.260$, $t = 8.313$, $p < 0.001$), perceived behavioral control ($\beta = 0.228$, $t = 6.952$, $p < 0.001$), and subjective norms ($\beta = 0.165$, $t = 5.108$, $p < 0.001$). Based on these results, one can conclude that COVID-19 impacts the antecedents of individuals' intention to start a business, showing strong evidence in support of H4a–c. The last hypothesis deals with the impact of COVID-19 on an individual's entrepreneurial intention. Table 5 shows the total effect of the COVID-19 pandemic on entrepreneurial intention, which is statistically significant ($\beta = 0.220$, $t = 6.492$, $p < 0.001$). In addition, this influence is positive, meaning that an increase in the values of the variable that measures COVID-19 leads to an increase in individuals' entrepreneurial intentions.

Figure 4 graphically illustrates the path analysis generated by SmartPLS 3.0. Note that the total effect is not plotted in this figure. Instead, the figure provides information on the inner model by showing the path coefficients along with their statistical significance (*t*-statistics).

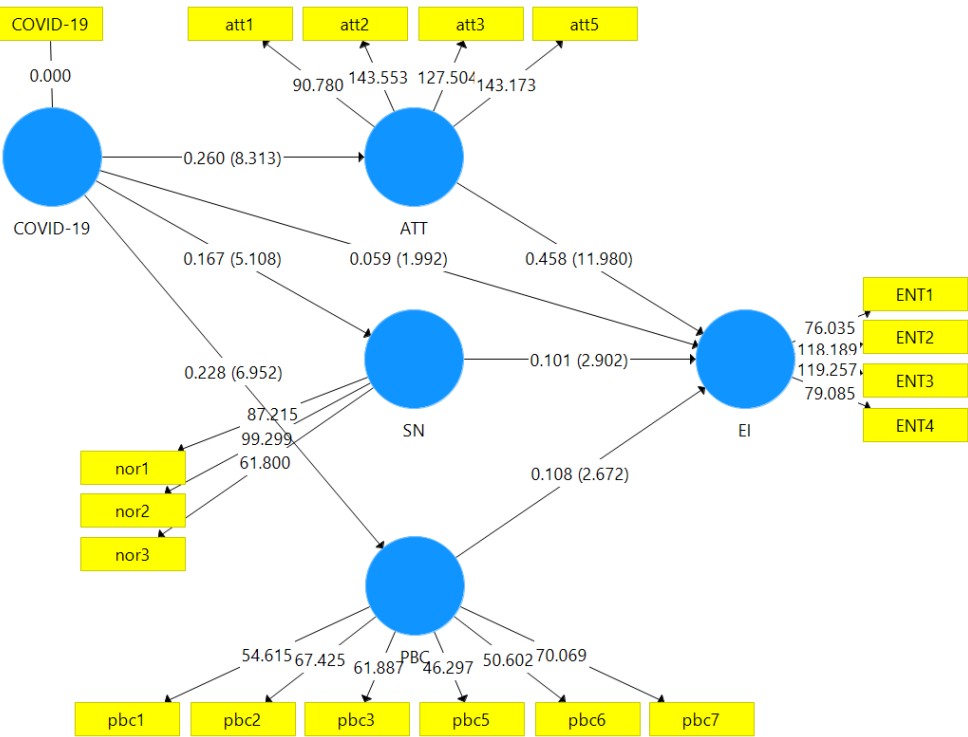

**Figure 4.** Path analysis—inner model: path coefficients (*t*-values). *Note*: ATT, attitude; EI, entrepreneurial intention; PBC, perceived behavioral control; SN, subjective norms; COVID-19, the COVID-19 pandemic.

## 5. Discussion

This paper aimed to examine the impact of the COVID-19 pandemic on individuals' intentions toward starting a business. The integration of two theories was proposed: the theory of planned behavior (Ajzen 1991), and the stimulus–organism–response perspective (Mehrabian and Russell 1974). The integration of these two theories offers a conceptual framework that can determine the impact of external stimuli (here represented by COVID-19) on entrepreneurial intention and its determinants.

The main finding of this work is that crisis, in addition to posing additional challenges to individuals and organizations, can also be seen as a generator of new opportunities. This finding is consistent with the limited research that has been conducted in this context (Ketchen and Craighead 2020; Krichen and Chaabouni 2021; Li et al. 2022; Ratten 2021; Usman and Sun 2022). Hence, due to the COVID-19 pandemic, individuals can find new business opportunities and a suitable situation to implement new ideas, which may lead to innovation (Brown and Rocha 2020). Such linkages can be seen with individuals' entrepreneurial behavior as well, including the intention to start a business. Thus, as this research demonstrates, entrepreneurial intention is positively affected by COVID-19 (seen as an opportunity). According to the findings of our work, individuals who perceive times of crisis as an opportunity may engage in startup activities to benefit from the situation, as their entrepreneurial intention is increased. This finding seems reasonable from the point of view of the entrepreneurial situation, which can form the perception of various risks that individuals face in a crisis context (Rayburn et al. 2022; Traczyk and Zaleskiewicz 2016). This is linked to the individuals' attitudes towards starting a business, which is an essential determinant of entrepreneurial intention and was found to be influenced by COVID-19. This result reinforces the positive impact that a crisis (seen as an opportunity and not as a threat) can have on entrepreneurial behavior, as shown in this study, which contradicts two prior studies (Godswill et al. 2021; Nguyen et al. 2020).

Nevertheless, Ruiz-Rosa et al. (2020) found similar results in a study on the social entrepreneurial intention of students from a university in Spain in the context of COVID-

19. Additionally, the data show that subjective norms and perceived behavioral control are positively influenced by COVID-19 which, in turn, affects entrepreneurial intention. These findings are consistent with the limited prior research carried out in the context of the COVID-19 pandemic (Botezat et al. 2022; Gomes et al. 2021; Nguyen et al. 2020; Ruiz-Rosa et al. 2020).

Such findings lead to the discussion on how to increase entrepreneurial activity. Various factors can influence entrepreneurial activity; however, one that all scholars agree on is that of education on entrepreneurship. Since entrepreneurship education has been found to be a significant determinant of individuals' intention towards engagement in startup activities (Çera et al. 2020; Dana et al. 2021; Durán-Sánchez et al. 2019; Hoppe 2016; Mwasalwiba 2010; Papagiannis 2018; Paray and Kumar 2020; Pedrini et al. 2017; Premand et al. 2016; Oo et al. 2018; Oosterbeek et al. 2010), it is unreasonable to doubt the role of education in this regard. Therefore, educational institutions are seen as critical actors in motivating students towards entrepreneurship since, through their curricula, they can be equipped with the knowledge and skills needed for starting and managing a business. Moreover, scholars claim that the entrepreneurial university environment is an essential factor that can increase entrepreneurial intention and actual behavior (García-Rodríguez et al. 2017; Ndou et al. 2018, 2019; Trif et al. 2022; Çera et al. 2021). Since COVID-19 has impacted the traditional means of providing entrepreneurship education (Hoti et al. 2022; Ndou 2021; Kripa et al. 2021), educational institutions should address the challenges and make use of innovative ways to deliver the best practices to equip students with adequate knowledge and skills (Cunningham 2022). Recently, there has been a discussion in the literature on the need to shift from the traditional means of offering entrepreneurship education to digital methods (Volkmann and Grünhagen 2022; Lehmann et al. 2022). This need to shift from the traditional approach to a new one is present due to COVID-19. Therefore, the COVID-19 pandemic has also created new challenges and opportunities for educational institutions.

## 6. Conclusions

### 6.1. Implications of the Study

Driven by the theory of planned behavior (Ajzen 1991) and the stimulus–organism–response perspective (Mehrabian and Russell 1974), this study provides a unique and improved research model for investigating the positive impact of COVID-19 on individuals' intentions to start a business in the context of three post-communist transition countries. Furthermore, the combination of these two theories provides the possibility of investigating the abovementioned relationship by seeing the COVID-19 pandemic as an external inducement (i.e., stimulus) that influences attitudes, subjective norms, perceived behavioral control, and entrepreneurial intention.

The findings of this research provide theoretical contributions and practical implications. Regarding this paper's contribution to the entrepreneurship literature, the authors believe that the integration of the two abovementioned theoretical lenses should be considered as a novelty of the paper. Putting the theory of planned behavior (Ajzen 1991) into a stimulus–organism–response paradigm (Mehrabian and Russell 1974) would be a useful approach that provides results. Therefore, this study adds to the existing literature by offering a new and unique conceptual framework, which may be useful for investigating the impacts of exogenous shocks on entrepreneurial intention and its determinants in a crisis context. In addition, in terms of theoretical contribution, the current paper demonstrates that a disaster or crisis that occurs, such as COVID-19, can not only pose additional challenges but also provide new opportunities which, in turn, lead to the increase in individuals' intentions towards starting a business. Therefore, our findings are valuable in strengthening the literature on entrepreneurial intention, which is ample in terms of research conducted in "normal times" but limited when a disaster or crisis occurs, such as the COVID-19 pandemic.

Regarding the practical implications of this research, from the policymakers' point of view, it is imperative to understand the effects of a crisis on individuals' intentions and

behavior toward startup activity, because this can lead to a reduction in unemployment—especially among young adults. Therefore, according to this research, policymakers and educational institutions should adjust the existing policies, strategies, instruments, and curricula to face the challenges raised by COVID-19 and benefit from the new opportunities.

*6.2. Limitations*

Although our research's goal was met, this study is not free of limitations. Firstly, the study focuses on individuals' intentions rather than their actual behavior toward starting a business. Even though there is a significant correlation between entrepreneurial intention and behavior (Bae et al. 2014; Joensuu-Salo et al. 2020), it is still not certain that intention will turn into behavior in either the near or far future (Bogatyreva et al. 2019). Secondly, from a methodological perspective, a crisis's impact should be measured by applying a pre- and post-test research design. Finally, the generalization of the findings obtained by the presented research model is limited to the countries that this study covers. Therefore, scholars should be advised to use and test the proposed conceptual framework in different contexts, as further research could contribute to overcoming the abovementioned limitations.

**Author Contributions:** Conceptualization, G.Ç. and I.D.; methodology, E.Ç. and G.Ç.; software, G.Ç.; validation, M.N. and E.Ç.; formal analysis, G.Ç.; investigation, E.Ç.; resources, E.Ç. and I.D; data curation, E.Ç.; writing—original draft preparation, G.Ç. and M.N.; writing—review and editing, M.N. and I.D.; visualization, G.Ç.; supervision, G.Ç. and I.D.; project administration, M.N.; funding acquisition, E.Ç. and I.D. All authors have read and agreed to the published version of the manuscript.

**Funding:** This research received no external funding.

**Institutional Review Board Statement:** Not applicable.

**Informed Consent Statement:** Not applicable.

**Data Availability Statement:** Not applicable.

**Conflicts of Interest:** The authors declare no conflict of interest.

## Appendix A

**Table A1.** Items and sources of the variables used in the research.

| Code | Items and Sources |
| --- | --- |
| Indicate your level of agreement with the following statements for each (1 = strongly disagree, 5 = strongly agree) | |
| | **COVID-19** (Krichen and Chaabouni 2021) |
| | The COVID-19 pandemic situation has made me optimistic in starting a business |
| | **Entrepreneurial intention** (Liñán and Chen 2006) |
| ei1 | I am ready to do anything to be an entrepreneur |
| ei2 | My professional goal is to become an entrepreneur |
| ei3 | I will make every effort to start and run my own firm |
| ei4 | I am determined to create a firm in the future |
| | **Attitude** (García-Rodríguez et al. 2017) |
| att1 | Being an entrepreneur implies more advantages than disadvantages to me |
| att2 | A career as entrepreneur is attractive for me |
| att3 | If I had the opportunity and resources, I would become an entrepreneur |
| att4 * | Being an entrepreneur would entail great satisfaction for me |
| att5 | Among various options, I would rather become an entrepreneur |

Table A1. *Cont.*

| Code | Items and Sources |
|------|-------------------|
| | **Perceived behaviour control** (García-Rodríguez et al. 2017) |
| pbc1 | I am usually able to protect my personal interests |
| pbc2 | When I make plans, I am almost certain to make them work |
| pbc3 | I can pretty much determine what will happen in my life |
| pbc4 * | For me, being an entrepreneur would be very easy |
| pbc5 | If I wanted to, I could easily pursue a career as entrepreneur |
| pbc6 | As entrepreneur, I would have complete control over the situation |
| pbc7 | As an entrepreneur, the chances of success would be very high |
| | **Subjective norms** (García-Rodríguez et al. 2017) |
| | Pursuing a career as an entrepreneur, how do people in your environment react? (1 = very negatively, 5 = very positively) |
| sn1 | Your close family |
| sn2 | Your friends |
| sn3 | Your fellow students/colleagues |

* Removed from the analysis due to the violation of the PLS-SEM assumptions.

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
