# Peer review of "Examining the Impact of COVID-19 on Entrepreneurial Intention through a Stimulus–Organism–Response Perspective"

_admsci, doi:10.3390/admsci12040184_

Round 1

Reviewer 1 Report

My only comment for the Authors is to review again the English. There are some grammar mistakes and not very well expressed ideas in some of the sentences. 

Author Response

Dear Editor,

Dear Reviewers,

We would like to thank you for your invitation to revise the manuscript. We are glad to be allowed to do so.

All valuable comments and suggestions have been strictly followed. The comments are quite convergent, and they were found very relevant to improving the paper. Many thanks for the constructive feedback.

As suggested by the Reviewers, we carefully revised the paper by:

  • Improving the introduction section by emphasizing the problem statement;
  • Bringing more clarity to the role of COVID-19 on the determinants of behavioral intention (based on the theory of planned behavior);
  • Enriching the Literature review with new and relevant sources;
  • Adding some new sub-sections and removing some others as suggested by the reviewers.

Kindly find all the changes in the revised manuscript, as the Track changes option in Microsoft Word has been applied.

Please find below the responses for each comment of the reviewers.

Yours sincerely,

Authors

Reviewer 2 Report

Thank you very much for the opportunity to review the article, please note the following:

1.     The article needs proofreading.

2.     Despite the authors having explained many things in the introduction section, I still cannot see the actual statement of the problem and the context of the study. The authors need to show the problem of the study in the introduction section. They should be able to explain the motives behind conducting this research. They should disclose where this research has been conducted, what was the problems there, and how can this research contribute to solving the problem of the study. The authors do not need to have many sections. Section 1.2 can be combined with the introduction and all is explained there.

3.     2.2 section should be renamed to the literature review and hypotheses development and then every hypothesis should have its own titles like 2.2.1 Attitude towards behavior and entrepreneurial intention and so on.

4.     The same thing should be applied to subjective norms; you should have a separate section with a separate title for subjective norms and entrepreneurial intention.

5.     Please include these recent articles as they are related to your research topic and they are very recent.

-        COVID-19 as an external enabler: The role of entrepreneurial self-efficacy and entrepreneurial orientation (2022).

-        Psychological features and entrepreneurial intention among Saudi small entrepreneurs during adverse times (2022).

-        Investigating the impact of institutions on small business creation among Saudi entrepreneurs (2022).

-        Exploring the influence of potential entrepreneurs’ personality traits on small venture creation: the case of Saudi Arabia (2022).

6.     The section measuring the impact of Covid 19 and entrepreneurial intention needs more empirical and theoretical support. I might agree with you to some extent that adverse situations should not be always looked at negatively as there might be always an opportunity there. Still, I am not able to see this in your arguments. Please discuss clearly how can Covid19 motivate people to start businesses, How can Covid19 change my attitude to start a business after the crisis? Your claim needs that you address the concept of resilience, as resilience will help people deal with challenges and setbacks effectively and get over them. Also, if we assume your assumption is true about the concept of attitude and Covid19. How about the SN, this concept is related to society, friends, and family, how can you say that Covid19 has changed your attitude and at the same time people’s attitudes and behavior? The same concerns the PBC, this one is again related to the resilience concept. Please read about entrepreneurial resilience (second article attached), the more resilience an individual has the more he is able to sustain a business or continue their business during the adverse time. Please read the articles I mentioned above and incorporate them in your article. You should have separate hypotheses for these three concepts. You cannot combine them together.

7.     You need to add a separate section for the implications of the study as this is very important to the study.

8.     You need to have a separate section for measures of the study to show the sources for the measures used in the study.

9.     As stated earlier, the problem of the study has been mentioned in the introduction section.

Round 2

Reviewer 2 Report

Satisfied